# Osteopontin Activation and Microcalcification in Venous Grafts Can Be Modulated by Dexamethasone

**DOI:** 10.3390/cells12222627

**Published:** 2023-11-15

**Authors:** Liam W. McQueen, Shameem S. Ladak, Georgia R. Layton, Kerry Wadey, Sarah J. George, Gianni D. Angelini, Gavin J. Murphy, Mustafa Zakkar

**Affiliations:** 1Department of Cardiovascular Sciences, Clinical Sciences Wing, Glenfield Hospital, University of Leicester, Leicester LE3 9QP, UK; 2Translational Health Sciences, Bristol Medical School, University of Bristol, Research Floor Level 7, Bristol Royal Infirmary, Bristol BS2 8HW, UK

**Keywords:** intimal hyperplasia, osteopontin, microcalcification, ^18^F-sodium fluoride, dexamethasone

## Abstract

Background: Osteopontin has been implicated in vascular calcification formation and vein graft intimal hyperplasia, and its expression can be triggered by pro-inflammatory activation of cells. The role of osteopontin and the temporal formation of microcalcification in vein grafts is poorly understood with a lack of understanding of the interaction between haemodynamic changes and the activation of osteopontin. Methods: We used a porcine model of vein interposition grafts, and human long saphenous veins exposed to ex vivo perfusion, to study the activation of osteopontin using polymerase chain reaction, immunostaining, and ^18^F-sodium fluoride autoradiography. Results: The porcine model showed that osteopontin is active in grafts within 1 week following surgery and demonstrated the presence of microcalcification. A brief pretreatment of long saphenous veins with dexamethasone can suppress osteopontin activation. Prolonged culture of veins after exposure to acute arterial haemodynamics resulted in the formation of microcalcification but this was suppressed by pretreatment with dexamethasone. ^18^F-sodium fluoride uptake was significantly increased as early as 1 week in both models, and the pretreatment of long saphenous veins with dexamethasone was able to abolish its uptake. Conclusions: Osteopontin is activated in vein grafts and is associated with microcalcification formation. A brief pretreatment of veins ex vivo with dexamethasone can suppress its activation and associated microcalcification.

## 1. Introduction

The long saphenous vein (LSV) remains the most frequently used conduit in coronary artery bypass grafting (CABG) [1]. However, the long-term success of CABG is limited by the development of intimal hyperplasia (IH) and accelerated atherosclerosis that leads to vein graft failure and major adverse events [2,3,4]. It is recognised that up to 40% of vein grafts occlude within 10 years of surgery, with estimated attrition rates of 1% to 2% per year between 1 and 6 years, and 4% per year between 6 and 10 years [5,6]. IH occurs because of chronic structural changes in vein grafts due to the abnormal migration and proliferation of smooth muscle cells (SMCs), accompanied by an increase in the amount of extracellular matrix (ECM) [7,8]. IH is a multifactorial process that starts during the harvesting of veins and is known to result in the activation of endothelial cells (EC) [7]. ECs are further activated by the sudden exposure to new mechanical forces in the arterial circulation, such as distension and shear stress, via the activation of different signalling pathways such as mitogen-activated protein kinase (MAPK) and nuclear factor-κB (NF-KB) [8,9,10]. Osteopontin (OPN) is a glycophosphoprotein that plays important roles in many physiological and pathophysiological processes [11]. Vascular smooth muscle cells (VSMCs) and ECs can make OPN at low levels; but at the site of injury such as after vein graft surgery, OPN can be overexpressed. Increased levels of OPN cause VSMCs to migrate and promote arterial wall remodelling, with this process regulated by of different pathways such as MAPK [12,13]. OPN is also upregulated at injury sites in porcine vein graft models [14,15]. Although calcification of atherosclerotic lesions is a typical feature of advanced stages of atherosclerosis in native arteries [16], it is also observed in human vein grafts [14,17]. However, the role of OPN and the temporal formation of microcalcification in vein grafts is poorly understood. Additionally, there is a lack of understanding of the interaction between haemodynamic changes and the activation of OPN. This work aims to assess the role of acute haemodynamic changes on OPN in the LSV ex vivo and associated microcalcification formation, with a focus on the role of a brief dexamethasone pretreatment of LSV upon modulating this process, considering that corticosteroids are recognised to inhibit OPN production [18].

## 2. Material and Methods

### 2.1. Reagents and Antibodies

Pharmacological inhibitors of P38 (Cell signaling technology, 5633S, Danvers, MA, USA), dexamethasone (Sigma-Aldrich, D2915, Dorset, UK), and antibodies (Table 1) were purchased commercially.

### 2.2. Comparative Reverse-Transcriptase Polymerase Chain Reaction

Total RNA was extracted using the RNeasy mini kit (Qiagen, Hilden, Germany, 74104). For mRNA studies, cDNA was synthesized from total RNA using the Tetro cDNA synthesis kit (Bioline, London, UK. BIO-65043). Gene expression was determined by quantitative real-time polymerase chain reaction (qRT-PCR) using SensiFast Probe Hi-ROX kit (Bioline, BIO-82005) and gene-specific primers (Table 2) on a Rotor gene Q (Qiagen) following the manufacturer’s protocol. Relative levels were calculated using the 2^−(ΔΔCt)^ method and mRNA expression was normalized to the housekeeping gene PPIA [19,20]. All primers were obtained from ThermoFisher Scientific.

### 2.3. Porcine Interposition Model

We used vein sections taken from previous animal interposition graft procedures that had been generated by the University of Bristol. These studies were conducted in line with the U.K. Home Office regulations (Animal Act 1986) and were undertaken under Project Licenses (PPL 30/3064 and PPL 30/2854) granted by the Home Office after formal review and approval by the University of Bristol Animal Welfare and Ethics Review Body (AWERB). These studies used female Yorkshire pigs (approx. 60 kg) receiving daily aspirin (300 mg) with food. The procedure was in line with established approaches by the group [21]. General anaesthesia was achieved with IV morphine (0.2 mg/kg) and propofol. Mechanical ventilation was maintained with isoflurane in oxygen/air. The activated clotting time (ACT) > 400 s was maintained with heparin (10,000–15,000 I.U.) and monitored every 15 min. During the surgical procedure, a continuous infusion of fentanyl (5 µg/kg/h) was administered along with 0.9% saline (4 mL/kg/h). After placement of soft vascular clamps, a 1.5 cm segment of the native carotid artery was excised and 1.8–2.00 cm and long saphenous vein grafts (diameter 4 ± 0.7 mm) were implanted via end-to-end anastomosis using continuous, running polypropylene 7–0 sutures (Prolene^®^, Ethicon, Somervillle, NJ, USA). Animals were kept for 2 weeks followed by termination under general anaesthesia and extraction of the previously implanted veins. For this study, we used *n* = 4 matched veins with *n* = 4 sections taken from each vein.

### 2.4. Ex Vivo Perfusion of Veins

We used our previously described model for ex vivo vessel perfusion [19,20]. Briefly, surplus segments of human LSV, resected during surgery from anonymized, consenting patients, were placed immediately in RPMI 1640 culture medium with 10% (*v*/*v*) foetal calf serum (FCS), 100 µg/mL penicillin, and 100 U/mL streptomycin. The study was approved under Leicester Biomedical Research Centre (BRICCS Ethics Ref: 09/H0406/114). Informed consent was obtained from all study participants prior to surgery and our use of human tissue conformed to the principles outlined in the Declaration of Helsinki. Sections of 6 cm were cut for use in ex vivo perfusion. LSV were either pretreated with dexamethasone (10 µM for 60 min) or remained untreated. Vein sections were cannulated with a 1/16” size male luer fitting (World Precision Instruments, Sarasota, FL, USA) and secured with a fine surgical tie. They were then mounted on a perfusion apparatus and exposed to arterial haemodynamics for different times or were maintained under static conditions as a control. Veins attached to the flow system were perfused under a mean arterial pressure of 65 mm Hg with M199 medium (containing 20% FCS, 100 U/mL penicillin, and 0.1 mg/mL streptomycin), which was oxygenated (oxygen content, 20 mL/L) and prewarmed to 37 °C. We targeted a wall shear stress of 12 ± 0.2 dyn/cm^2^ and adjusted flow rates based upon the diameter of the cannulae. Shear stress was calculated using the equation: τ = 4µǪ/πr^3^, where τ represents shear stress; µ, viscosity; Ǫ, flow rate; π, pi; r, radius; and assuming flow is through a non-deformable cylinder and is laminar; and considering the viscosity of water at 37 °C.

### 2.5. LSV Sections Immunostaining

For immunostaining of fresh frozen tissues, slides were incubated in pre-chilled 10% neutral buffered formalin (Sigma Aldrich, HT5014-1CS) for 15 min at 4 °C. Thereafter, slides were dehydrated in ethanol (50–100%) and allowed to dry. Sections were then incubated in immunofluorescence blocking buffer (Cell Signalling Technology, 12411) for 30 min at room temperature, followed by overnight incubation in primary antibody (Table 1) and immunofluorescence antibody dilution buffer (Cell Signalling Technology, 12378) at 4 °C. Secondary antibody was applied for one hour followed by nuclei staining with DAPI (Invitrogen, ThermoFisher Scientific, D1306). Control slides were routinely stained in parallel by substituting IgG, or the specific IgG isotype. All images were acquired using either a Zeiss Axioscope with AxioVision V4.3 software or a Zeiss LSM 510 UV laser scanning confocal microscope (Carl Zeiss GmBH. Oberkochen, Germany). The expression of proteins was assessed by quantification of fluorescence intensity for multiple cells as reported previously [19,20].

### 2.6. RNAscope In Situ Hybridisation

Frozen vein sections were processed for fluorescent in situ hybridization by RNAscope according to the manufacturer’s guidelines (Bio-Techne Ltd., Abingdon, UK). The gene examined in the vein sections was OPN (RNAscope^®^ Probe—Hs-SNAI1-C2, catalogue number 560421-C2) and hybridization was performed using the RNAscope^®^ Multiplex Fluorescent Reagent Kit v2 (Bio-Techne Ltd., 323100). These sections were co-stained with CD31 or α-SMA antibodies using an RNAscope^®^ Multiplex Fluorescent v2 Assay combined with the Immunofluorescence–Integrated Co-Detection Workflow (Bio-Techne Ltd.). OPN puncta were counted using the cell-counter plugin in Fiji [22]. After the assay was performed, dots quantification was performed based on the average number of dots per cell as previously described [23].

### 2.7. Sample Preparation and Histological Examination

Vein tissue samples were frozen and embedded in cryo-embedding medium (OCT, CellPath, Newtown, UK, 6478.1) on dry ice. The frozen block was placed in an air-tight container at −80 °C prior to sectioning. Furthermore, the blocks were equilibrated to −20 °C in a cryostat ~1 h. OCT embedded veins were sectioned into 10 μm slices and mounted onto SuperFrost^®^ Plus slides (Fisher Scientific, Loughborough, UK, 12312148). In all cases, images were generated using an Aperio Slide Scanner using ImageScope software version 12.3.3 (Leica Biosystems, Wetzlar, Germany). Histological examination was performed using Von Kossa and Alizarin staining to assess for the presence of calcification.

### 2.8. ^18^F-Sodium Fluoride Autoradiography

Tissue was prepared based on a previously published protocol [24]. Briefly, formalin-fixed paraffin-processed sections were rehydrated and equilibrated in phosphate-buffered saline (PBS) for 30 min. Sections were then incubated with 100 kBq/mL of ^18^F-sodium fluoride in PBS for 1 h at room temperature with a blocking control (10 µmol/L sodium fluoride) before two 5 min washes: one in PBS and one in deionized water. Dried sections were exposed to a high-resolution autoradiography plate (BAS-IP-SR 2040; Cytiva, Westborough, MA, USA), which were imaged on an autoradiography imager (Amersham Typhoon IP Biomolecular Imager, Cytiva).

### 2.9. LSV Culture Well

We used the modified culture well model as described previously [25]. Briefly, the vein was placed in wash medium (20 mm Hepes-buffered RPMI 1640 supplemented with 2 mm l-glutamine, 8 mg/mL gentamycin, 100 IU/mL penicillin, and 100 mg/mL streptomycin). Segments were opened longitudinally and cut transversely into three 5–10 mm segments. Vein segments were cultured separately with the endothelial surface uppermost, in culture medium (RPMI 1640 supplemented with 30% foetal calf serum, 2 mm L-glutamine, 100 IU/mL penicillin, and 100 mg/mL streptomycin) using a modification of the method of Pederson and Bowyer as described previously. Culture media were changed every 2 days.

### 2.10. EC Culture and Shear Stress

Pooled human umbilical vein endothelial cells (HUVECs) were obtained from PromoCell (Heidelberg, Germany. C-12203) and were cultured to full confluency on glass microscope slides precoated with 1% (*v*/*v*) gelatin (Sigma-Aldrich, G1393). HUVECs were then exposed to laminar, unidirectional shear stress for varying times, using parallel plate flow chambers, or were maintained in static conditions. Briefly, the glass slides were placed into the parallel plate chambers and sealed with a silicon sheet gasket. A reservoir containing 30 mL RPMI 1640 culture medium (ThermoFisher Scientific, 11875093), supplemented with 2% (*v*/*v*) FCS, 100 µg/mL penicillin, and 100 U/mL streptomycin attached to a closed-circuit loop of silicon tubing (VWR, Radnor, PA, USA and Elkay, Basingstoke, UK) was connected to the chambers. HUVECs were then cultured at 37 °C and 5% CO_2,_ and shear stress applied using a multi-channel peristaltic pump (Watson-Marlow, Marlow, UK) [19,20,26].

### 2.11. Immunocytochemistry

HUVECs were fixed and stained as previously reported [19,20,26] using pre-chilled 10% neutral buffered formalin (Sigma Aldrich, HT5014-1CS) for 15 min at 4 °C. Thereafter, slides were dehydrated in ethanol (50–100%) and allowed to dry. Cells were incubated in immunofluorescence blocking buffer (Cell Signaling Technology, 12411) for 30 min at room temperature followed by overnight primary antibody incubation at 4 °C and AlexaFluor fluorophore-conjugated secondary antibody for 1 h at room temperature. Immunoglobulin matched controls at the same concentration as the primary antibodies were used as staining-specific negative controls. Nuclei were labelled with DAPI and coverslips were mounted with ProLong^®^ Gold Antifade Reagent (ThermoFisher Scientific, P36930). Slides were visualized using a Zeiss Axio Observer Z1 inverted microscope and quantification of fluorescence intensity was measured as integrated intensity in whole cells divided by the total number of cells using CellProfiler 2.0.

### 2.12. Statistical Analysis

All experiments were performed using *n* = 4. For experiments where only two groups were analysed, data were subjected to a paired, two-tailed *t*-test. For experiments where more than two groups were analysed, a one- or two-way ANOVA was used depending on the number of independent variables, followed by post-hoc pairwise comparisons with Bonferroni correction for multiple comparisons. If datasets were large enough, normal distribution was assessed with the D’Agostino-Pearson test; all data that were assessed passed normality tests, and as such, parametric analyses were appropriate. The cut-off value for statistical significance was 0.05. Data are presented as the mean ± SD. All statistical analysis was performed with GraphPad Prism 7.0 as previously reported [19,20].

## 3. Results

### 3.1. OPN and Microcalcification Activation in Porcine Venous Grafts

We initially investigated the activation of OPN and associated markers of calcification formation in a porcine model of interposition grafts. Our results showed a significant upregulation of OPN protein expression at 7 and 14 days post-surgery compared to the control (Figure 1A). This was associated with increased Von Kossa (Figure 1B) and Alizarin (Figure 1C) staining in the tissue at both time points indicating the presence of microcalcification. Moreover, using autoradiography we noted a significant increase in ^18^F-sodium fluoride uptake at 7 and 14 days, further confirming the presence of microcalcification (Figure 1D).

### 3.2. Acute Arterial Haemodynamics Are Associated with OPN Activation

To further understand the temporal activation of OPN and the role of acute changes in haemodynamics within veins once exposed to arterial flow when implanted into arterial circulation, we exposed surplus segments of human LSV to acute arterial flow for 4 h.

We noted using RT-PCR that acute arterial flow is associated with the activation of pro-inflammatory responses such as IL-8 and MCP-1 as previously described [19,20] (Figure 2A). Additionally, using both RT-PCR (Figure 2B) and RNAscope (Figure 2C), we observed that acute arterial flow was associated with both the activation of OPN at the RNA level, and with increased protein expression as observed with immunostaining (Figure 2D).

Considering that P38-MAPK is known to regulate the activation of OPN, we investigated the impact upon OPN expression of suppressing P38 activity using a known P38 inhibitor, and noted that the pretreatment of LSV segments with a P38 inhibitor resulted in a significant reduction in IL-8, MCP-1 (Figure 2A), and OPN expression using RT-PCR (Figure 2B) and RNAscope (Figure 2C). This was also associated with a significant reduction in OPN protein expression as noted by immunostaining (Figure 2D).

### 3.3. Dexamethasone Role in Suppressing OPN Expression

We reasoned that dexamethasone, which is known to modulate P38-MAPK activity, can suppress OPN expression. We noted that the pretreatment of LSV with dexamethasone suppressed MCP-1 and IL-8 transcripts (Figure 3A) and P38 activation (phosphorylation) using histo-immunostaining (Figure 3B). Dexamethasone pretreatment also suppressed OPN expression both at the RNA level using RT-PCR (Figure 3C) and RNAscope (Figure 3D), as well as at protein level (Figure 3E). Similar results were noted in vitro, where the exposure of HUVECs to acute arterial shear stress resulted in the activation of OPN expression but this was modulated by the P38 inhibitor and dexamethasone pretreatment (Figure 3F,G).

### 3.4. Prolonged LSV Culture after Exposure to Acute Arterial Haemodynamics Is Associated with Microcalcification Formation

Considering that the formation of microcalcification in tissue can require a prolonged time for onset, we exposed segments of LSV to acute arterial flow with or without dexamethasone pretreatment for 4 h and went on to culture these segments for up to 14 days using the validated culture well model. We noted that culturing LSV for 7 and 14 days after exposure to acute arterial haemodynamics is associated with a significant increase in OPN protein expression (Figure 4A), as well as an increased presence of microcalcification markers identified using Von Kossa (Figure 4B) and Alizarin staining (Figure 4C). This was significantly suppressed in the dexamethasone pretreated LSV segments. Moreover, using autoradiography, we noted a significant ^18^F-sodium fluoride uptake in the cultured LSV segments at 7 and 14 days which was abolished in the dexamethasone pretreated segments (Figure 4D). Additionally, there was significant suppression of pro-inflammatory responses (IL-8 and MCP-1) (Figure 4E) noted by PCR, suggesting that a single brief dexamethasone pretreatment of LSV is capable of regulating OPN-associated microcalcification formation.

## 4. Discussion

The LSV remains the most used conduit for CABG which can be due to the ease of harvest, ability to obtain long length for multiple grafting, and because clinical trials using arterial grafting have not produced consistent results demonstrating long-term superiority [27,28]. The use of LSV is complicated by high rates of restenosis or occlusion, broadly termed vein graft failure, due to the development of IH which is a complex process that is triggered almost immediately once the vein is harvested and is related to the activation of multitude of inflammatory pathways [7,19,20].

Over the years, many attempts been made to modulate IH with limited success of any single strategy [29]. Veins in situ of the coronary circulation are known to experience different haemodynamics to arteries, and once a vein is implanted into the arterial circulation it is suddenly exposed to new forces such as changes in shear stress and increased flow and pressure which activates ECs and triggers an inflammatory process [8,9,10,30].

Additionally, the impact of vein graft failure extends beyond cardiac surgery and this similar process of IH can impact the patency of veins used in vascular surgery and in patients requiring fistula formation for dialysis. This makes any attempt to better understand and modulate IH in vein grafts important to large groups of patients [31,32,33].

We have demonstrated previously that the exposure of LSV to arterial haemodynamics can activate pro-inflammatory responses mediated by the MAPK and NF-KB pathways. We also showed that a brief pretreatment of veins with dexamethasone can effectively modulate P38-MAPK activation and reduce the inflammatory response within venous ECs following acute changes in haemodynamics [20].

OPN is known to play a role in the development of IH in animal studies and is thought to be implicated in the regulation of SMC activity. However, no previous studies have looked at the role of acute changes in haemodynamics upon the activation of OPN and the resultant microcalcification within vein grafts [12,14,15,34]. A study of hypertension within an animal model showed that the treatment with a P38 MAPK inhibitor can significantly reduce OPN protein expression in target organs, suggesting that enhanced OPN expression is reflective of end-organ damage in hypertension, and that its suppression therefore correlates with end-organ protection [13]. Similarly, the inhibition of P38 MAPK prevented the hypoxia-induced increase in OPN in mesangial cells [35]. Moreover, OPN expression in the lungs is increased in a murine model of allergen-induced chronic airway disease, suggesting an important role in airway remodelling [36,37]. The use of glucocorticoids such as dexamethasone has also demonstrated favourable effects in animal models of allergic asthma by suppressing OPN activity [18].

In this study, we confirmed and expanded upon previous work by showing that OPN is active in porcine interposition grafts and have demonstrated the presence of microcalcification using Von Kossa and Alizarin staining. Next, we showed that the suppression of P38 activation using a known P38 inhibitor as a pretreatment can suppress OPN in segments of veins exposed to acute arterial haemodynamics. Similarly, a brief pretreatment of LSV with dexamethasone was able to suppress OPN activation. Moreover, we demonstrated that the prolonged culture of veins following exposure to acute arterial haemodynamics can lead to the formation of microcalcification but that this can be suppressed by a single brief pretreatment with dexamethasone. Finally, we showed that ^18^F-sodium fluoride uptake was significantly increased as early as 1 week after grafting in the porcine interposition graft model, with similar results obtained in our post-flow culture model, with a brief single pretreatment of LSV with dexamethasone being able to abolish this uptake at 1 and 2 weeks.

In addition to existing evidence linking OPN/microcalcification to ^18^F-sodium fluoride uptake in coronary arteries [24], OPN has been gaining more interest as a biomarker for increased risk of myocardial infarction (MI) in asymptomatic patients, as well as its role as a predictor of adverse outcomes in patients with carotid stenosis [38,39,40,41,42]. Limited work has been conducted in patients undergoing CABG, with one study showing decreased levels 72 h after surgery [43]; however, the authors did not study level changes at later time points considering evidence that OPN is detected at later times after injury [44]. Another study of pre-operative OPN levels in patients with stable CAD undergoing elective CABG using cardio-pulmonary bypass found that OPN is higher in patients with prior MI and in patients taking exogenous insulin. Interestingly, OPN did not vary in relation to CAD severity or the left ventricular ejection fraction but was elevated along with an increased higher EuroScore [45].

Many previous studies have looked at ^18^F-sodium fluoride uptake in coronary arteries and its co-localisation with OPN and microcalcification [46,47,48,49]; however, very limited worked has addressed this in LSV grafts. One study demonstrated a low uptake of ^18^F-sodium fluoride in vein grafts, but this study was focused on native coronary vessels and looked at a late time points for imaging (12 months), rather than earlier stages like our results showing clear activation of OPN/microcalcification and ^18^F-sodium fluoride uptake [47]. This suggests that imaging vein grafts for ^18^F-sodium fluoride uptake at earlier time-points after surgery, coupled with serial measurements of serum OPN levels, may potentially improve the role of OPN in predicting the risk of vascular inflammation and later, IH in vein grafts.

One of the limiting factors for a successful treatment of IH has been the availability of an easy, readily available therapeutic that will not interfere with the process of surgery and will have broad effects without systemic side effects. A brief topical pretreatment of the LSV with dexamethasone can provide a perfect solution as we have demonstrated it is able to modulate vascular inflammation and OPN expression; both of which are linked to the development of IH.

Our study is limited as we did not study the co-localization of OPN/microcalcification and ^18^F-sodium fluoride uptake because the animal model was not designed to investigate OPN expression or the impact of dexamethasone treatment, and this study was conducted between three units based upon locally available assays. We were also limited by the logistics of the bioreactor for LSV culture over prolonged time periods (weeks) due to the need to frequently change the circulating medium and the resultant potential change in the environment/temperature the veins are cultured at, or by needing to stop and restart the flow during a single prolonged experiment. It will be interesting to study OPN as a biomarker within patients undergoing surgeries using LSV grafts, and to perform ^18^F-sodium fluoride CT-PET scans early after surgery. Doing so will offer the opportunity to link the levels of OPN expression and ^18^F-sodium fluoride uptake to later IH formation. It will also be interesting to study the role of a brief dexamethasone pretreatment on the above process in vivo.

## 5. Conclusions

We conclude that OPN is activated within vein grafts in response to acute exposure to arterial haemodynamics ex vivo and that a brief pretreatment of veins with dexamethasone can suppress both its activation and associated microcalcification, which may modulate the development of intimal hyperplasia in vein grafts. Furthermore, the demonstrated ^18^F-sodium fluoride uptake early both in vivo and ex vivo in relation to OPN and microcalcification formation may provide a potential non-invasive modality to image such changes early in patients following surgery as a precursor for inflammation and intimal hyperplasia. This will allow more tailored therapeutics to be developed and studied without the need to carry out invasive imaging in patients.

## Figures and Tables

**Figure 1 cells-12-02627-f001:**
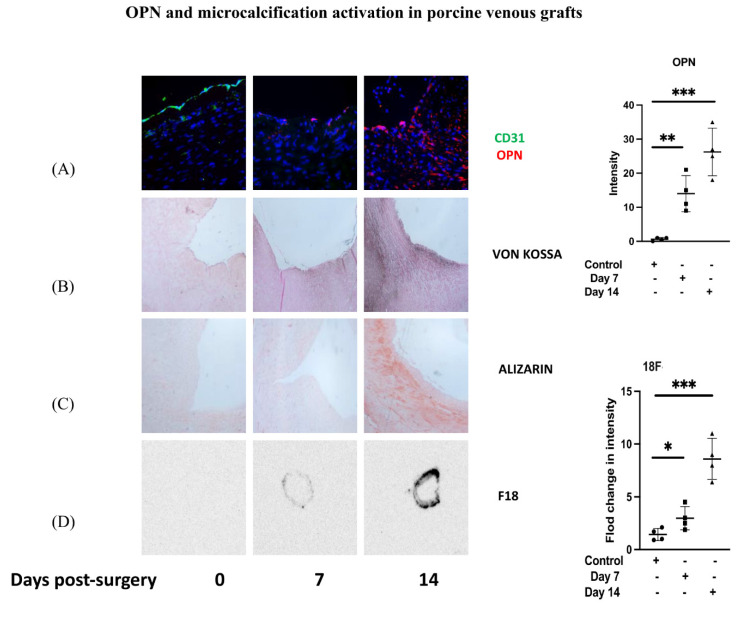
Porcine interposition vein grafts at 7 and 14 days post-surgery were obtained and compared to the control. (**A**) Immunofluorescence staining with specific antibody for OPN (red) and CD31 (green) as an EC marker was carried out. OPN was quantified in sections and averaged for each experimental group. Representative images, values from four independent experiments, and mean values are shown. (**B**) Representative images for Von Kossa staining of sections at 7 and 14 days post-surgery compared to the control showing increased stain uptake. (**C**) Representative images for Alizarin staining of sections at 7 and 14 days post-surgery compared to the control, showing increased stain update. (**D**) ^18^F-sodium fluoride uptake at 7 and 14 days post-surgery compared to the control. Representative images, values from four independent experiments, and mean values are shown. *p*-value * < 0.05, ** < 0.01, *** < 0.001.

**Figure 2 cells-12-02627-f002:**
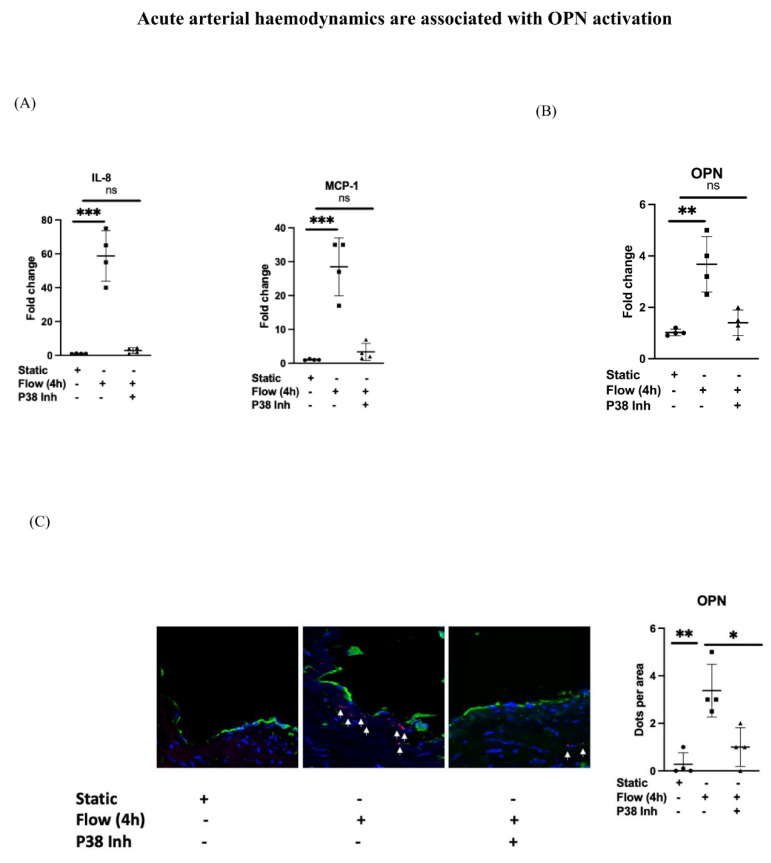
LSV were either pretreated with P38 inhibitor for 60 min or remained untreated. They were then mounted on a perfusion apparatus and exposed to LSS for 4 h. (**A**) Transcript levels of MCP-1 and IL-8 were measured by comparative RT-PCR. Values from four independent experiments and mean values are shown. (**B**) Transcript levels of OPN were measured by comparative RT-PCR. Values from four independent experiments and mean values are shown. (**C**) RNAscope with a probe specific for the OPN gene (arrows indication expression of genes) quantified in multiple sections and averaged for each experimental group. Representative images, values from four independent experiments, and mean values are shown. (**D**) Immunofluorescence staining with specific antibody for OPN (red) and CD31 (green) as an EC marker was carried out with and without P38 inhibitor pretreatment. OPN was quantified in sections and averaged for each experimental group. Representative images, values from four independent experiments, and mean values are shown. *p*-value * < 0.05, ** < 0.01, *** < 0.001. ns = not significant.

**Figure 3 cells-12-02627-f003:**
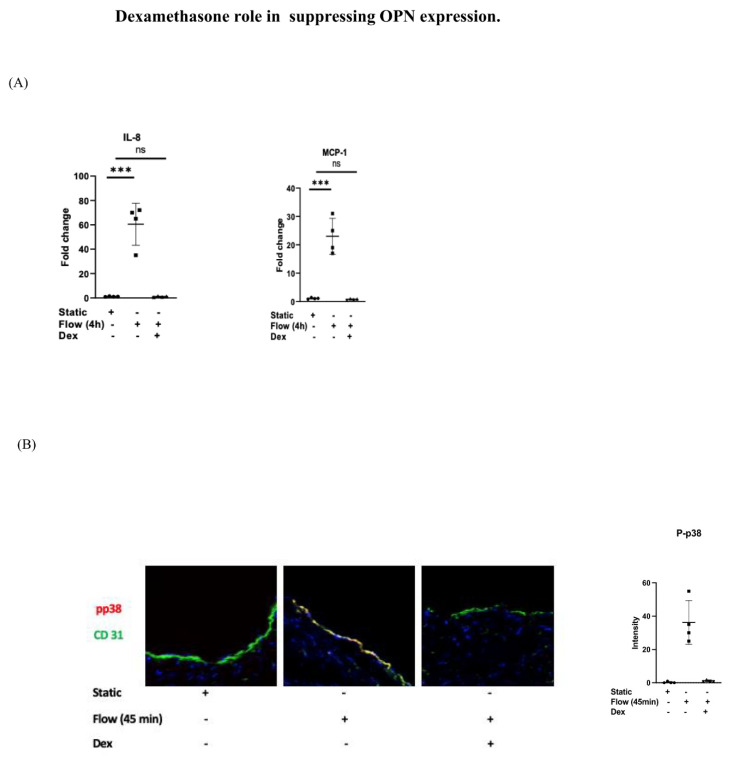
LSV were either pretreated with dexamethasone for 60 min or remained untreated. They were then mounted on a perfusion apparatus and exposed to LSS for 4 h or 45 min. (**A**) Transcript levels of MCP-1 and IL-8 were measured by comparative RT-PCR. Values from four independent experiments with veins either pretreated with dexamethasone or kept untreated as control and mean values are shown. (**B**) Immunofluorescence staining with specific antibody for active (phospho) P38 (red) and CD31 (green) as an EC marker was carried out with and without dexamethasone pretreatment. (**C**) Transcript levels of OPN were measured by comparative RT-PCR. Values from four independent experiments with veins either pretreated with dexamethasone or kept untreated as a control and mean values are shown. (**D**) RNAscope with a probe specific for the OPN gene (arrows indication expression of genes) quantified in multiple sections and averaged for each experimental group. Representative images, values from four independent experiments, and mean values are shown. (**E**) Immunofluorescence staining with specific antibody for OPN (red) and CD31 (green) as an EC marker was carried out with and without dexamethasone pretreatment. OPN was quantified in sections and averaged for each experimental group. Representative images, values from four independent experiments, and mean values are shown. (**F**) HUVECs were either pretreated with dexamethasone or a P38 inhibitor for 60 min or remained untreated. They were then mounted on a perfusion apparatus and exposed to LSS for 4 h and transcript levels of OPN were measured by comparative RT-PCR. Values from four independent experiments and mean values are shown. (**G**) HUVECs were either pretreated with dexamethasone or P38 inhibitor for 60 min or remained untreated. They were then mounted on a perfusion apparatus and exposed to LSS for 4 h and immunofluorescence staining with specific antibody for OPN (red) and CD31 (green) as an EC marker was carried out with and without a P38 inhibitor or dexamethasone pretreatment. OPN was quantified in sections and averaged for each experimental group. Representative images, values from four independent experiments, and mean values are shown. *p*-value * < 0.05, ** < 0.01, *** < 0.001, ns = not significant.

**Figure 4 cells-12-02627-f004:**
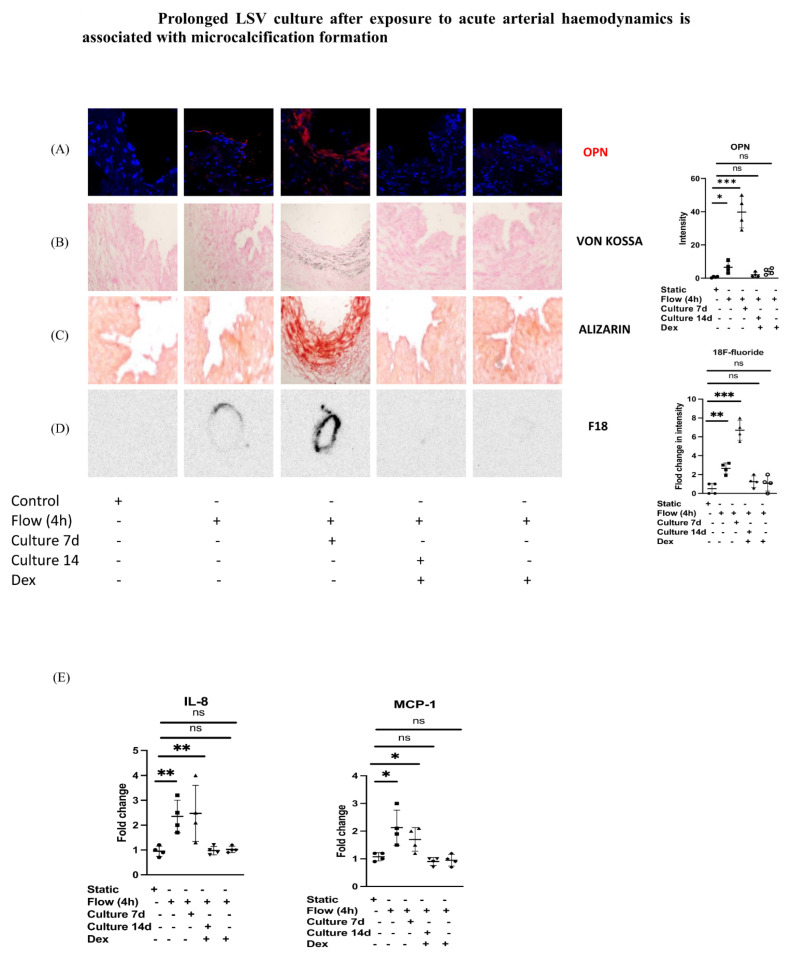
LSV were either pretreated with dexamethasone (10 µM for 60 min) or remained untreated. They were then mounted on a perfusion apparatus and exposed to LSS for 4 h then cultured in a culture well for 7 and 14 days. (**A**) Immunofluorescence staining with specific antibody for OPN (red) and CD31 (green) as an EC marker was carried. OPN was quantified in sections and averaged for each experimental group. Representative images, values from four independent experiments, and mean values are shown. (**B**) Representative images for Von Kossa staining of sections at 7 and 14 days post-surgery compared to the control showing increased stain uptake. (**C**) Representative images for Alizarin staining of sections at 7 and 14 days post-surgery compared to the control showing increased stain uptake. (**D**) ^18^F-sodium fluoride uptake at 7 and 14 days post-surgery compared to the control. Representative images, values from four independent experiments, and mean values are shown. (**E**) Transcript levels of MCP-1 and IL-8 were measured by comparative RT-PCR. Values from four independent experiments with veins either pretreated with dexamethasone or kept untreated as the control and mean values are shown. *p*-value * < 0.05, ** < 0.01, *** < 0.001, ns = not significant.

**Table 1 cells-12-02627-t001:** Antibodies.

Antibodies	Manufacturer	Product Code
Anti-CD31 (PECAM1)	R&D Systems (Abingdon, UK)	BBA7
Anti-αSMA	Antibodies.com (Cambridge, UK)	A82445
Anti-Phospho-P38 MAPK	Cell Signaling Technology (Danvers, MA, USA)	4511
Anti-Mouse Alexa Fluor™ 488	ThermoFisher Scientific (Bleiswijk, The Netherlands)	A-11001
Anti-Rabbit, Alexa Fluor™ 488	ThermoFisher Scientific	A-11008
Anti-Mouse Alexa Fluor™ 568	ThermoFisher Scientific	A11031
Anti-Goat Alexa Fluor™ 594	ThermoFisher Scientific	A11058
Anti-Rabbit, Alexa Fluor™ 568	ThermoFisheScientific	A11011
Osteopontin	Biolegend (San Diego, CA, USA)	691302
Alizarin Red S	Sigma-Aldrich (Dorset, UK)	TMS-008-C
Von Kossa Stain	Abcam (Cambridge, UK)	ab150687

**Table 2 cells-12-02627-t002:** Primer list.

Primers	Manufacturer	Assay ID
PPIA	ThermoFisher Scientific	Hs99999904_m1
OPN	ThermoFisher Scientific	Hs00959010_m1
MCP-1	ThermoFisher Scientific	Hs00234140_m1
IL-8	ThermoFisher Scientific	Hs00174103_m1

## Data Availability

The data presented in this study are available on request from the corresponding author.

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
