# Peer review of "Osteopontin Activation and Microcalcification in Venous Grafts Can Be Modulated by Dexamethasone"

_cells, 2023, doi:10.3390/cells12222627_

Round 1

Reviewer 1 Report

Comments and Suggestions for Authors

Interesting study that contributes to intimal hyperplasia prevention after Cardiac/Vascular Surgery. Following are my suggestions to improve the manuscript.

Title: I would say Osteopontin instead of OPN.

Abstract: I would say Osteopontin instead of OPN, since you do not spare words using OPN. Furthermore, possibly, it is better not to use acronyms in the abstract section. Or, at least, explain the meaning of the acronym the first time you use it.

Line 13: “Intimal” should become “intimal”. Furthermore, I think for “it is” you mean “its”. 

Line 20: explain the meaning of LSV. Furthermore, world-wide this vein is called great saphenous vein (GSV), not long saphenous vein.

If you need, you can spare words in the abstract section: for instance, vein graft can be abbreviated VG.

Line 31-2: “Coronary” should become “coronary”.

Line 43: explain the meaning of MAPK and NF-KB: likely, these acronyms are familiar for you, but not for all the potential readers of your manuscript.

Lines 62-3 (and following): add the city and state of each manufacturer.

Line 84: even if you send the reader to your previous publication; you should report some basic data (number of pigs sacrificed, number of specimens analized, etc.).

Line 283: normalize the font to the rest of the manuscript.

Comments on the Quality of English Language

Minor editing of English language required.

Author Response

Dear reviewer

Thank you for your time and valuable comments

we agree with all the comments and we changed the manuscript accordingly as indicated in the revised version in red.

Regarding your comments about the name. The vein in known as either long saphenous or great saphenous vein with large body of literature in pubmed using this term. Our published work based on being UK based always used this term and we would like to continue that however we added that its called some time great saphenous vein 

I hope you find the revised version satisfactory.

Yours sincerely 

Mustafa Zakkar

Reviewer 2 Report

Comments and Suggestions for Authors

The manuscript is interesting and has practical value, but needs major correction.

1.     Page 1 line 13: the word “intimal” must not with a capital letter.

2.     Page 1 line 19: Replace “OPN is active grafts” with “OPN is active in grafts”

3.     Page 1 line 31-34: The problem of IH is not only in coronary surgery. It is also relevant in lower extremity arterial surgery, especially  for the by-pass below the knee.

4.     Page 2 line 43: Abbreviation MAPK need clarification (MAPK- mitogen-activated protein kinase)

5.     Page 3 line 89: Specify which vein was used for interposition and what was the average diameter of the venous grafts. Also specify please, the anastomosis were interrupted or continuous. 

6.     Page 4 line 158: Replace “he vein” with “the vein”

7.     Page 5 line 203: A parenthesis is missing before the Figure 1A.

8.     The authors describe the technique of the pre-treatment of LSV with Dexamethasone in the caption of Figure 1. I think it would be better to add it to the section -Materials and methods as well. 

9.     In the discussion, the authors focus mainly on coronary surgery. The same problem is acute in the surgery of arteries below the knee. If the authors agree with me that the mechanism of  IH is identical, then it is better to briefly discuss this study in the second aspect as well. 

Also, the materials and methods do not specify the number of experimental animals. This is important data for the article.

Author Response

Dear reviewer

Thank you for your time and valuable input

we agree with the points raised and corrected the manuscript accordingly as indicated in red.

we added the required information and abbreviations. we added the n number and information in methods as well as more details about the size of vein.

We agree that IH is not only unique to cardiac surgery but by virtue of being a cardiac surgeon and our work focuses on cardiac patients, that is why this was more oriented to CABG.  We have added a section in discussion about IH being a problem for other surgeries.

We hope that you find the revised version satisfactory

Yours sincerely 

Mustafa Zakkar